# Assessment of Radiation Risk Perception and Interest in Tritiated Water among Returnees to and Evacuees from Tomioka Town within 20 km of the Fukushima Daiichi Nuclear Power Plant

**DOI:** 10.3390/ijerph20032690

**Published:** 2023-02-02

**Authors:** Xu Xiao, Hitomi Matsunaga, Makiko Orita, Yuya Kashiwazaki, Yasuyuki Taira, Thu Zar Win, Jacques Lochard, Thierry Schneider, Noboru Takamura

**Affiliations:** 1Department of Global Health, Medicine and Welfare, Atomic Bomb Disease Institute, Nagasaki University, 1-12-4 Sakamoto, Nagasaki 852-8523, Japan; 2Department of Global Health, Medicine and Welfare, Atomic Bomb Disease Institute, Nagasaki University Graduate School of Biomedical Sciences, 1-12-4 Sakamoto, Nagasaki 852-8523, Japan; 3Nuclear Protection Evaluation Center (CEPN), 92260 Fontenay-aux-Roses, France

**Keywords:** Fukushima Daiichi Nuclear Power Plant accident, risk perception, tritiated water, returnee, evacuee

## Abstract

In this study, we aimed to investigate radiation risk perception, mental health, and interest in tritiated water among evacuees from and returnees to Tomioka town, Japan, as well as to evaluate the intention to return (ITR) among evacuees living inside and outside Fukushima Prefecture. Of the 1728 respondents, 318 (18.4%) and 1203 (69.6%) participants reported living outside and inside Fukushima Prefecture, and 207 (12.0%) reported living in Tomioka. The ITR was not significantly different between those who lived inside and outside the prefecture among the evacuees. Similarly, there were no significant differences in radiation risk perception, mental health, and interest in tritiated water. However, the evacuees were independently associated with a motivation to learn about tritiated water (OR = 1.242, 95%Cl: 1.041–1.438, *p* = 0.016), reluctance to consume food from Tomioka (OR = 1.635, 95%Cl: 1.372–1.948, *p* < 0.001), and concern that adverse health effects would occur because of the Fukushima Daiichi Nuclear Power Plant accident (OR = 1.279, 95%Cl: 1.055–1.550, *p* = 0.012) compared to returnees, according to logistic regression analysis. Interestingly, the returnees were found to have better mental health but lower life satisfaction than the evacuees. These findings suggest the importance of ongoing risk communication about radiation exposure and tritiated water among residents regardless of their place of residency.

## 1. Introduction

At 14:46 on 11 March 2011, a massive earthquake struck Fukushima Prefecture on the Pacific coast of Japan, leading to a tsunami that flooded the Fukushima Daiichi Nuclear Power Plant (FDNPP) and destroyed its electrical facilities, resulting in a serious accident involving meltdowns. As a result of this nuclear accident, radionuclides were released into the atmosphere [1]. Subsequently, the Japanese government issued evacuation orders, leading to a total of 164,865 residents evacuating their homes, moving elsewhere in or even outside Fukushima Prefecture. As of 2022 February, 27,789 people were still living as evacuees (6392 inside and 21,392 outside Fukushima Prefecture; 5 people’s statuses were unknown) [2]. Furthermore, reconstruction was progressing slowly in towns that were uninhabited for a long time, and support for basic living needs, such as medical care, education, public transportation, and job opportunities, was still lacking [3]. In addition, the evacuees’ intention to return (ITR) to their hometowns was found to be affected by anxiety and their perception of risk regarding the health effects of radiation exposure [4].

Risk perception refers to the subjective judgement that people make about the characteristics and severity of a risk, which may be influenced by a wide range of psychological, social, institutional, and cultural factors. Numerous studies have shown that gender, age, race, education, region of residence, cultural origin, social status, income, knowledge of risks, experience of disasters, and styles of communication strategies all have an impact on perceptions [5,6,7,8,9]. They also show that there is often a gap between subjective risk perceptions and objective facts.

The public’s assessment of risk is often different from the judgments made by experts, which are based on data and their knowledge of the risk [10]. Slovic et al. argued that the perception of risk associated with nuclear energy has its own specificity and its characteristics that induce a higher perception of environmental risk [11]. Nuclear knowledge is highly specialized, and nuclear accidents are rare; thus, compared to other natural or technical disasters, the perception of risk associated with such accidents is more likely to be detached from objective facts.

A reasonable perception of risk, as close as possible to the objective level of risk, can help individuals protect themselves effectively against risky situations. Conversely, too low a perception of risk can cause people to ignore dangers, leading to serious consequences, while too high a perception level can lead to emotional and behavioral overreactions and lead to mental health disorders. A World Health Organization report on the health effects of the Chernobyl nuclear accident, carried out 20 years after the accident, suggested that the most serious public health problem caused by the nuclear accident was poor mental health [12]. It is also recognized that people who are forced to evacuate after an accident have poor mental health. Therefore, it is essential to monitor the mental health of victims of nuclear accidents in order to provide them with adequate and efficient support.

Effective risk communication can help people improve their perception of risk when it differs significantly from objective facts. To do this, it is necessary to constantly research and study the factors that influence the public’s perception of risks in order to better target communication.

Since the Fukushima nuclear accident, our research team has worked on radiation risk communication to help reduce anxiety about radiation and support evacuees’ decisions to return to their hometowns. In this study, we focused on the town of Tomioka, which is located within 20 km of the Fukushima Daiichi nuclear power plant (Figure 1).

Evacuation orders were issued for Tomioka on 12 March 2011. Six years after the accident, in April 2017, these evacuation orders were lifted, except in respect of the difficult-to-return zone, which accounted for about 20% of the town. According to data from 2018, Tomioka completed the decontamination of 418,583 houses, 11,958 public facilities, and 18,841 km of roads in preparation for the return of its residents, resulting in a significant decrease in its environmental radiation levels [13]. Radiation dose rates were calculated to be 1.4 mSv/y in 2018 and 1.1 mSv/y in 2019 (excluding the difficult-to-return zone) [14]; the latter value, significantly, amounts to less than half of the average global natural radiation dose of 2.4 mSv/y. However, as of December 1, 2022, 9679 evacuees (7853 inside Fukushima and 1826 outside Fukushima) had not yet returned to Tomioka, and only 2078 residents were living in Tomioka [15].

At present, the Japanese government and Tokyo Electric Power Company are considering releasing water containing trace amounts of radionuclides such as tritium (hereafter referred to as tritiated water) produced during the decommissioning process into the Pacific Ocean. Tomioka has a seaport that began operating after the accident. Therefore, its residents likely have considerable concerns about the environmental and social effects of tritiated water and its release into the ocean.

It is necessary to help residents who have suffered as a result of nuclear accidents to establish good perceptions of nuclear radiation risks. The purpose of this study was to examine and compare differences in various factors, such as radiation risk perception, mental health, and interest in tritiated water, between Tomioka evacuees and returnees and between evacuees living inside and outside Fukushima Prefecture, to achieve a better understanding of risk communication.

In addition, as studies on the perceived risks of nuclear accidents among evacuees who have actually been exposed to nuclear disasters are, to date, limited, the results of this study may have some extrapolation value.

## 2. Materials and Methods

### 2.1. Participants

In November 2021 and January 2022, we distributed 9655 questionnaires to residents aged > 18 years who had resident cards for Tomioka on 11 March 2011 and still had them in August 2021, including new residents who started living in Tomioka after the accident. We received a total of 2899 responses, and after eliminating those with missing values, inconsistent responses, and 12 new residents who had started living in Tomioka after the accident, a total of 1728 samples were left for inclusion in the study. All methods and procedures in this study were reviewed and approved by the Nagasaki University Graduate School of Biomedical Sciences Ethics Committee (No. 21082702).

Of the 1728 respondents, 1521 (88.0/%) residents with household registration in Tomioka had not yet returned (living outside Tomioka); we classified these as evacuees and assigned them to Group 1. Within this group, 318 (18.4%) and 1203 (69.6%) participants were living outside (classified into Group 1a) and inside (classified into Group 1b) Fukushima Prefecture, respectively. A total of 207(12.0%) residents had returned to live in Tomioka; we classified these as returnees and assigned them to Group 2.

### 2.2. Questionnaire Design

The questionnaire used in the study was a continuation of those previously used by our research team for studies in Kawauchi village, Tomioka town, and Okuma town, with appropriate adjustments made based on the research results. The sociodemographic characteristics of the study participants included sex, age, employment, marital status, and living with a child under 18 years of age. Regarding the evacuees’ ITR to Tomioka, we asked whether they decided not to return, were undecided, or wanted to return. We also asked the residents about issues related to anxiety about internal radiation exposure by inquiring whether they were reluctant to consume food from Tomioka. To evaluate the participants’ perceived radiation health risk, we asked whether they thought adverse health or genetic effects would occur because of the FDNPP accident. Furthermore, we asked whether they were motivated to learn more basic knowledge about radiation and tritiated water. The Japanese version of the six-item Kessler Psychological Distress Scale (K6), which was validated in previous reports [16], and life satisfaction were used to assess mental health and psychological distress status. Life satisfaction was evaluated by the question “Are you satisfied with your current life?”, with the responses “yes”, “probably yes”, “probably no”, and “no” provided as options. We measured nonspecific mental health distress as a primary outcome using the K6. In the K6, the participants were asked whether they had experienced any of the following symptoms in the previous 30 days: feeling so sad that nothing could cheer them up; feeling nervous, hopeless, restless, or fidgety; feeling that everything was an effort; feeling worthless. Each question was scored on a five-point Likert scale from 0 (none of the time) to 4 (all of the time), with higher scores signifying worse mental health status (range: 0–24). The cutoff point for predicting severe mental illness was 13 points [17]. Furthermore, we asked whether the participants had considered consulting a professional regarding radiation and whether they knew of a place where they could consult radiation professionals (“yes” or “no” were provided as options for both items).

### 2.3. Statistical Analysis

The chi-squared test was used to compare differences in each factor depending on the area of residence, including Tomioka. First, to clarify the relationship between other factors and the distance to the disaster area factor, we compared the factors of residents living outside (1a) and inside (1b) Fukushima in Group 1. Responses of “yes” and “probably yes” and of “no” and “probably no” on a four-point Likert scale were classified as “yes” and “no” in the analysis, respectively. To confirm the differences between evacuees and returnees, we also compared Group 1 and Group 2 using the chi-squared test. Then, factors that independently differed among groups were further investigated using the logistic regression analysis of each group, with Group 2 used as a reference. Statistical analysis was performed using IBM SPSS Statistics (version 25; SPSS Japan, Tokyo, Japan), and all *p*-values < 0.05 were considered statistically significant.

## 3. Results

Table 1 shows the participants’ sociodemographic characteristics and the results of the chi-squared test for all groups. First, we compared the residences of evacuees outside (1a) and inside (excluding Tomioka) (1b) of Fukushima to determine whether the distance from the disaster area in Tomioka to a participant’s present residence affected each factor. No differences in ITR were found between those with residency inside and outside of Fukushima. Moreover, no significant differences were found between the groups outside and inside of Fukushima, except for employment (yes; outside 18.2% vs. inside 25.9%, *p* = 0.005) and living with children (yes; outside 10.7% vs. inside 15.5%, *p* = 0.031). These findings indicate that the participants’ ITR, risk perceptions about radiation health effects, anxiety about radiation exposure, interest in tritiated water, and mental health did not depend on the distance of their residences from the affected area in Tomioka.

To investigate the differences between returnees and evacuees, we compared Group 1 with Group 2. The results indicated that there were more males in Group 2 than in Group 1 (male; 53.6% vs. 44.2%, *p* = 0.011). In addition, Group 2 contained more participants ≥ 60 years and not living with a child (age < 18 years) than Group 1 (≥60; 84.1% vs. 75.2%, respectively, *p* = 0.005; no; 93.2% vs. 85.5%, respectively, *p* = 0.002). The marital status and employment factors were not significantly different between Groups 1 and 2. Regarding the mental health indicators, the K6 found no significant differences between Groups 1 and 2, except for satisfaction with one’s current life (yes; 49.8% vs. 32.9%, respectively, *p* < 0.001). Evacuees were found to be more satisfied with their current lives than returnees to Tomioka. Regarding anxiety about radiation exposure, those in Group 2 reported feeling less reluctant to consume food from Tomioka compared with those in Group 1 (yes; 35.3% vs. 57.7%, respectively, *p* < 0.001). Furthermore, Group 2 reported feeling less worried than Group 1 about the health effects of radiation caused by the FDNPP accident (yes; 50.2% vs. 60.0%, respectively, *p* < 0.001). However, 54.3% of the respondents thought that genetic effects might occur because of the FDNPP accident, although no significant difference in risk perception about genetic effects was found between the groups. Compared to Group 1, Group 2 contained a significantly higher percentage of participants who had considered consulting a professional regarding radiation, as well as of those who knew of a place where they could consult a radiation professional (yes; 18.8% vs. 13.8%, respectively, *p* = 0.037; yes; 48.3% vs. 36.7%, respectively, *p* < 0.001, respectively). The results of the chi-squared test revealed no significant differences in motivation to learn more basic knowledge about radiation or tritiated water between the groups.

Table 2 shows the results of the multivariate logistic regression analysis as a reference for the returnees (Group 2). The variance inflation factors (VIF) of all the variables were much lower than 10 (all VIF values were between 1 and 2), and there was no serious multicollinearity. The K6 indicator of depression was independently associated with the evacuees (Group 1), and indigenous residents with higher K6 scores were more likely to still live outside of Tomioka (odds ratio (OR) = 1.855, 95% confidence interval (CI): 1.121–3.071, *p* = 0.016). However, they were more likely to be satisfied with their current lives (OR = 2.013, 95%Cl: 1.650–2.456, *p* < 0.001). The evacuees were more likely to be reluctant to consume food from Tomioka (OR = 1.635, 95%Cl: 1.372–1.948, *p* < 0.001) and more likely to be concerned that adverse health effects would occur because of the FDNPP accident (OR = 1.279, 95%Cl: 1.055–1.550, *p* = 0.012). However, they may have been more likely to have lower motivation to learn about tritiated water (OR = 1.242, 95%Cl: 1.041–1.438, *p* = 0.016) and less likely to consider consulting a professional regarding radiation (OR = 1.855, 95%Cl: 1.198–2.870, *p* = 0.006), and fewer knew of a place where they could consult such a radiation professional (OR = 1.451, 95%Cl: 1.067–1.973, *p* = 0.017).

## 4. Discussion

### 4.1. Sex and Age Distribution of Returnees and Evacuees

The present results show that the proportion of males was higher in the returnees group than in the evacuees group. Furthermore, a higher percentage of returnees were ≥60 years of age. A previous study in Tomioka regarding ITR reported that older males had higher ITR than younger females and males [18]. Further, in a study on risk perceptions related to the Fukushima accident, Japanese men tended to report having lower risk perceptions of the health effects of the Fukushima nuclear accident [19]. In addition, the likelihood of returning to one’s hometown has been reported to increase after one reaches the legal retirement age, especially among men [20]. By contrast, the willingness of working adults to return to their hometowns has been shown to be very low [21]. For retired seniors, the lack of job opportunities in Tomioka had little impact on their ITR; thus, life support, including social and medical care, is likely to be a more important consideration regarding their ITR. Gender may also have an impact on ITR, as males were found to be more likely to want to return than females [22]. In summary, it is important to provide more adequate risk communication to young people and women to improve their risk perceptions; it is also important to provide more life and work support to younger groups.

### 4.2. Returnees and Evacuees Living with Children under Age of 18 Years

Compared to evacuees, a smaller percentage of returnees were found to be living with children under the age of 18 years. By contrast, no significant difference was found between those living with children inside and outside of Fukushima. A survey conducted in 2020 on residents of Tomioka found that higher proportions of those who were hesitant to return and did not want to return to Tomioka were living with children under the age of 18 years compared to those who had returned. In studies on the Fukushima accident, it was also reported that the perception of risk was higher among those living with children and grandchildren [19]. The characteristics of population movement, in general, show a tendency toward people moving to regions with high education levels, good economic conditions, good living facilities, and high population numbers [23]. Since the FDNPP disaster, Tomioka has not yet been rehabilitated, and its local living, educational, and medical facilities all remain inadequate [24]. This may also make families with teenagers more reluctant to return.

### 4.3. Motivation to Learn about Tritiated Water among the Tomioka Residents

In this study, approximately 70% of residents reported wanting to know more about tritiated water. However, the motivation levels were not significantly different among those living in areas such as inside Tomioka, inside Fukushima but excluding Tomioka, and outside Fukushima. The Response to Radiological Disasters Committee of the Japan Society for Radiological Research, which has described the possible health effects of tritium in a way that is easily understood by the public and that is based on sound scientific evidence, has lent its support to the decision to discharge Advanced-Liquid-Processing-System-purified water containing tritium generated and stored at the FDNPP into the Pacific Ocean [25]. However, in contrast to concerns about the health effects of discharging tritiated water, a survey in May 2021 reported that the most common concern among the respondents about discharging tritiated water into the ocean was the occurrence of new reputational damage (40.9%), followed by prejudice and discrimination against citizens (18.1%), and the decline of industry in the prefecture (12.1%). However, only 11.0% of people were concerned about “health damage” [26]. Our present results suggest the importance of continuing risk communication about radiation exposure and tritiated water to residents regardless of where they live. There is also a need to unite the efforts of all sectors of society in a sustained effort to improve national and international reputations.

### 4.4. Impact of ITR and Distance to Residential Area on Radiation Risk Perception and Anxiety

#### 4.4.1. Impact of Distance on Risk Perception

The evacuees had a higher risk perception of radiation health effects than the returnees. However, there were no significant differences between the residents living inside and outside of Fukushima. A previous study found that residents within 500 m of a nuclear power plant had the lowest risk perception, whereas those within 1.4 km had higher risk perception [27]. Other studies have shown that risk perceptions are lowest among nuclear power plant staff and residents within 5 km of a plant, are higher among villagers 5–10 km from a plant, and are strongest among residents 10–20 km from a plant [28]. One study found that residents of Tokyo and Osaka had higher perceptions of nuclear risk than residents of Fukushima who had not experienced evacuation [19]. These findings are consistent with those of the present study, which may be because people living around nuclear power plants actively or passively obtain more information related to nuclear power and are more aware of nuclear safety, thereby lowering their risk perception.

#### 4.4.2. Risk Perception and the Social Environment

It has been reported that, when there is less knowledge about risk and more access to information, the perception of risk and fear of the overall impact of a risk is higher [29]. In the information age, the public has a variety of channels they can use to obtain information related to nuclear radiation; however, the credibility of such information can be mixed, with biased or even incorrect information often spreading widely. Incomplete and inaccurate public information about the consequences of nuclear accidents and disposal measures used several years after such accidents, as well as inconsistencies in international wind assessments, may lead to the public’s risk perception of nuclear radiation being more likely to be negative. Some studies about risk perception have found that people with higher trust in government regulations tend to have lower levels of perceived environmental risk [30]. Findings regarding the Fukushima accident suggest that risk perception is significantly lower among people who trust the central government as a source of information than among those who do not [19]. Since the evacuation order was lifted, multiple teams of experts have continued to communicate risks to Fukushima residents, and those who have returned to Tomioka have had considerable access to accurate information provided by professionals. People who have not yet returned to Tomioka have had less access to expert consultation services; thus, their trust in statements on the TV/radio and from friends and nuclear radiation non-experts, as well as their distrust of government authorities, contribute positively to their perception of high risk. Therefore, the perception of risk among Fukushima evacuees may increase dynamically over time. Further, science manuals and targeted leaflets provided by nuclear radiation experts may be more effective in reducing their risk perceptions than other methods.

#### 4.4.3. Impact of Evacuation and Accident Experience on Risk Perception

Evacuations themselves can also cause elevated risk perception. One study showed that the risk perception of Fukushima residents who experienced mandatory evacuation was similar to that of those who voluntarily evacuated. By contrast, Fukushima residents who did not experience mandatory evacuation had lower levels of risk perception than residents of Osaka and Tokyo [19]. This suggests that disaster-related experiences, including the ongoing evacuation itself, may enhance people’s perceptions of risk. In addition, traumatic experiences elevate risk perception.

### 4.5. Evacuees Feel More Anxious about Eating Food Produced in Tomioka

Fukushima Prefecture has set standard values for the radioactive content of various types of food [31]. The committed effective doses of radiation from local foods were calculated as being between 19 and 74 μSv/y for children and 39 and 100 μSv/y for adults during 2018 and 2019, well below the regulatory limit (<1 mSv/y). Radio cesium concentrations in wild edible plants and mushrooms were found to be relatively higher than those in vegetables and fruits [14], but their contribution to committed effective doses remains limited. Even with continued daily consumption of such food items for more than a year, the potential for health problems is low [32]. Returnees can avoid unnecessary internal exposure to radio cesium by following the food recommendations of authorities. The International Health Organization has previously reported that not only thyroid cancer but also other cancers caused by radiation will not increase in Fukushima Prefecture in the future [33].

In this study, people who lived outside of Tomioka were found to be more concerned about food safety than those who lived in Tomioka. Regarding our finding that evacuees were more likely to not know where to access nuclear radiation consultation sites, this may be because returnees were given more information about food security risks in Tomioka. In April 2019, a food inspection station was set up near Tomioka City Hall to measure the concentration of radioactive substances contained in ordinary food and to eliminate residents’ doubts and anxiety about nuclear radiation. Residents can ask the inspection station to test the amount of nuclear radiation in food and issue a report for free. In addition, individual consultations are available with nuclear radiation experts. Residents can borrow the portable radiometer from the food inspection station for free or invite staff to visit their homes to check the radiation doses in their environment. The food inspection station also regularly publishes food radiation dose monitoring results on the Tomioka homepage. As a result, residents who have returned to Tomioka are able to make informed choices about consuming foods they believe to be safe and may therefore be less anxious about food, as a result of the aforementioned scientific services and counseling provided by experts from food inspection stations.

### 4.6. Returnees Are More Likely to Consult with Specialists

We found that levels of motivation to acquire basic nuclear radiation knowledge were not significantly different between evacuees and returnees, suggesting that evacuees were not indifferent to nuclear radiation information. We suggest that the observed difference in the “desire to consult specialists” results from the greater number of risk communication services available in Tomioka, and that returnees are more aware of how these consultations with specialists are carried out, because they have already benefited from them several times. We will study the experiences of people who received expert advice in future studies. We recommend that more information about the radiological risk be provided to people outside of Tomioka.

### 4.7. Life Satisfaction among Returnees and Evacuees

The residents who returned to Tomioka were found to have lower life satisfaction than the evacuees. Considering that the returnees had lower risk perception and better mental health, their dissatisfaction might have been due to the incomplete reconstruction of Tomioka, as well as the town’s poor basic living facilities and transportation system [34,35]. The timing of their return may also have had an impact on their life satisfaction. A previous study on Kawauchi village showed that evacuated residents who returned in the early phase after the accident had a high sense of satisfaction with their current life and lower risk-perception a decade after the FNDPP accident [36]. The evacuation order for Tomioka was lifted in 2017, 6 years after the accident. This may have been one of the reasons for the low levels of life satisfaction seen among the returning residents included in this study. In addition, expectations for the revival of Tomioka may have been a confounding factor in terms of life satisfaction.

### 4.8. Mental Health among Returnees and Evacuees

One of the more interesting findings of this study was that the returnees had better mental health (lower K6 scores) than the evacuees, even though they also had lower life satisfaction. This result is consistent with previous studies, which reported that evacuees had lower levels of mental health than returnees [37]. Unexpected evacuations tend to worsen the mental health of evacuees [38]. Factors such as their being forced to move to unfamiliar places, separation from family and friends, and difficulties in rebuilding their lives may impair the mental health of evacuees. Moreover, regarding long-term evacuations, in addition to ongoing adverse psychological effects, evacuation-related health effects, such as lifestyle-related illnesses, also tend to worsen mental health. The return of residents to their homes can be impacted by many issues, such as delays in restoring their living environments and measures to reduce the effects of radiation [39], which may cause disagreement among family members; this difference has been found to be greater among residents far from Fukushima [40]. The excessive perception of risks associated with radiation can cause psychological stress [41]. A reasonable level of risk perception can help individuals protect themselves effectively in the face of hazardous situations, but conversely, too high a level of risk perception can cause individuals to have excessive emotional and behavioral reactions, which can worsen mental health. The results of the present study show that the risk perception level of residents of Tomioka who remained evacuated was higher than that of returning residents, which may have been one of the reasons why their mental health levels were lower than those of returning residents.

### 4.9. Limitations

This study had some limitations. First, fewer residents had returned to Tomioka than had remained evacuated; thus, there may have been bias when comparisons were made between the two. Second, this was a cross-sectional study; thus, it was not possible to verify the causal relationships between place of residence, risk perception, and life status. In addition, Tomioka includes areas that were difficult to return to before the evacuation order was lifted, which may have led to bias regarding the residential areas and different intentions of residents to return to their places of origin. Third, we examined differences in the level of interest in tritiated water among residents of Tomioka according to their residential area and whether they had returned to their homes. Since this is an issue of great concern in respect of the use of treated FDNPP water worldwide, detailed analysis of the issue is required in future studies. Fourth, our measure of life satisfaction was limited to respondents’ current life satisfaction, as we did not have specific data on the explanatory factors of risk perception, or well still the living conditions of these people. Although life satisfaction was not the main objective of our study, more relevant research will be undertaken in the future given its importance in the post-nuclear accident context.

## 5. Conclusions

In this study, it was found that those returning to Tomioka were more likely to be male, older, and not living with children. People in Tomioka were also found to be more likely to be dissatisfied with their lives, to have better mental health, and to have less anxiety about eating food from the area. They were also more willing to talk to radiation experts and to know where to consult experts on radiation-related issues. Meanwhile, evacuees living outside of Tomioka were more aware of and anxious about the risks of radiation. Approximately 70% of the respondents reported wanting to know about tritiated water, and no significant differences were found among those living in the following areas: inside Tomioka, inside Fukushima but excluding Tomioka, and outside Fukushima. In addition, no significant difference in ITR was found between those living inside and outside of Fukushima. To promote more basic knowledge on radiation and improve understanding of the health effects of radiation among people living outside of Tomioka, more risk communication services may need to be provided to them. Particular attention should be paid to females, young people, and families living with children. In addition, more risk communication services on tritiated water should be provided to residents, regardless of where they are living. Finally, these findings suggest that the Japanese government should pay more attention to the construction and revitalization of basic living facilities in Tomioka to improve life satisfaction among its residents.

## Figures and Tables

**Figure 1 ijerph-20-02690-f001:**
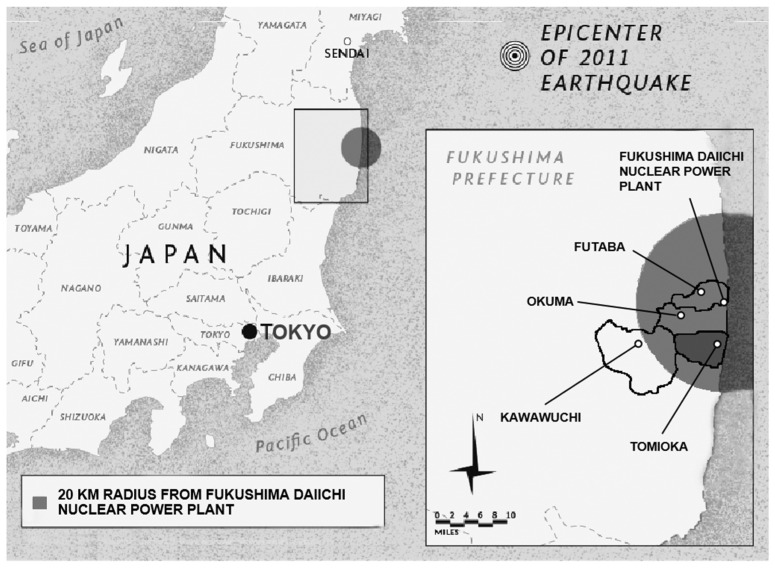
Map of the Fukushima prefecture showing the 20 km radius of the Fukushima Daiichi Nuclear Power Plant.

**Table 1 ijerph-20-02690-t001:** Sociodemographic characteristics of the study participants by area of residence.

Variables	Reference	Overall % (n) N = 1728	Group 1		*p*-Value (1a vs. 1b)	Group 2	*p*-Value (Group 1 vs. Group 2)
			Outside Fukushima ^1a^ % (n) 318 (18.4)	Inside Fukushima ^1b^ % (n) 1203 (69.6)		Living in Tomioka % (n) 207 (12.0)	
Sex	Female Male	54.7 (945) 45.3 (783)	56.3 (179) 43.7 (139)	55.7 (670) 44.3 (533)	0.899	46.4 (96) 53.6 (111)	0.011 *
Age (years)	<60 ≥60	23.7 (410) 76.3 (1318)	22.3 (71) 77.7 (247)	25.4 (306) 74.6 (897)	0.274	15.9 (33) 84.1 (174)	0.005 *
	<70 (including <60) ≥70	49.8 (860) 50.2 (868)	51.3 (163) 48.7 (155)	51.3 (617) 48.7 (586)	0.512	38.6 (80) 61.4 (127)	<0.001 *
Employment	Yes No	24.2 (418) 75.8 (1310)	18.2 (58) 81.8 (260)	25.9 (311) 74.1 (892)	0.005 *	23.7 (49) 76.3 (158)	0.931
Marriage	Yes No	71.5 (1235) 28.5 (493)	70.8 (225) 29.2 (93)	70.9 (853) 29.1 (350)	0.504	75.8 (157) 24.2 (50)	0.331
Living with a child (age < 18 years)	Yes No	13.6 (235) 86.4 (1493)	10.7 (34) 89.3 (284)	15.5 (187) 84.5 (1016)	0.031 *	6.8 (14) 93.2 (193)	0.002 *
Intention to return	Decided not to return Undecided Want to return	60.7 (923) 27.7 (422) 11.6 (176)	61.6 (196) 29.9 (95) 8.5 (27)	60.4 (727) 27.2 (327) 12.4 (149)	0.134	-	-
Satisfaction with current life	Yes Probably yes Probably no No	7.1 (123) 40.7 (703) 37.1 (641) 15.1 (261)	11.6 (37) 40.9 (130) 33.0 (105) 14.5 (46)	6.7 (80) 42.5 (511) 37.7 (453) 13.2 (159)	0.285	2.9 (6) 30.0 (62) 40.1 (83) 27.1 (56)	<0.001 *
K6	High ≥ 13 Low < 13	14.0 (242) 86.0 (1486)	13.5 (43) 86.5 (275)	14.6 (176) 85.4 (1027)	0.654	11.1 (23) 88.9 (184)	0.240
Reluctant to consume food from Tomioka	Yes Probably yes Probably no No	25.9 (448) 29.1 (503) 32.1 (554) 12.9 (223)	31.8 (101) 26.1 (83) 30.2 (96) 11.9 (38)	26.4 (317) 31.3 (377) 30.8 (370) 11.6 (139)	0.504	14.5 (30) 20.8 (43) 42.5 (88) 22.2 (46)	<0.001 *
Adverse health effects will occur because of the FDNPP accident	Yes Probably yes Probably no No	26.4 (457) 32.3 (559) 33.0 (570) 8.2 (142)	33.6 (107) 29.2 (93) 29.6 (94) 7.5 (24)	25.5 (307) 33.7 (405) 33.7 (406) 7.1 (85)	0.247	20.8 (43) 29.5 (61) 33.8 (70) 15.9 (33)	<0.001 *
Genetic effects will occur because of the FDNPP accident	Yes Probably yes Probably no No	21.2 (366) 33.2 (573) 36.5 (631) 9.1 (158)	24.8 (79) 32.1 (102) 34.3 (109) 8.8 (28)	20.6 (248) 33.7 (406) 37.6 (452) 8.1 (97)	0.447	18.8 (39) 31.4 (65) 33.8 (70) 15.9 (33)	0.234
Motivated to learn more basic knowledge about radiation	Yes Probably yes Probably no No	19.8 (342) 39.4 (680) 31.3 (540) 9.6 (166)	21.4 (68) 37.4 (119) 29.2 (93) 11.9 (38)	18.4 (221) 40.4 (486) 32.3 (388) 9.0 (108)	0.522	25.6 (53) 36.2 (75) 28.5 (59) 9.7 (20)	0.408
Motivated to learn about tritiated water	Yes Probably yes Probably no No	33.0 (570) 37.4 (647) 22.2 (383) 7.4 (128)	31.0 (105) 38.4 (122) 19.2 (61) 9.4 (30)	31.8 (382) 38.1 (458) 22.9 (276) 7.2 (87)	0.630	40.1 (83) 32.4 (67) 22.2 (46) 5.3 (11)	0.275
Considered consulting a professional regarding radiation	Yes No	14.4 (249) 85.6 (1479)	14.8 (47) 85.2 (271)	13.5 (163) 86.5 (1040)	0.584	18.8 (39) 81.2 (168)	0.037*
Knew of a place to consult radiation professionals	Yes No	38.1 (658) 61.9 (1070)	35.5 (113) 64.5 (205)	37.0 (445) 63.0 (758)	0.648	48.3 (100) 51.7 (107)	<0.001 *

Note: chi-squared test. Group 1: evacuees; Group 2: returnees; (1a): evacuees living outside Fukushima; (1b): evacuees living inside Fukushima (excluding Tomioka); K6: 6-item Kessler Psychological Distress Scale; FDNPP: Fukushima Daiichi Nuclear Power Plant. *: significant difference using the chi-squared test.

**Table 2 ijerph-20-02690-t002:** Factors independently associated with evacuees compared to returnees among Tomioka residents.

Variables	Unit	OR (95% Cl)	*p*-Value
K6	≥13/<13	1.855 (1.121–3.071)	0.016 *
Satisfaction with current life	Yes/No	2.013 (1.650–2.456)	<0.001 *
Motivated to learn about tritiated water	No/Yes	1.242 (1.041–1.438)	0.016 *
Reluctant to consume food from Tomioka	Yes/No	1.635 (1.372–1.948)	<0.001 *
Adverse health effects will occur because of FDNPP accident	Yes/No	1.279 (1.055–1.550)	0.012 *
Considered consulting a professional regarding radiation	No/Yes	1.855 (1.198–2.870)	0.017 *
Knew of a place to consult radiation professionals	No/Yes	1.451 (1.067–1.973)	0.006 *

Note: Logistic regression analysis. OR: odds ratio; CI: confidence interval; K6: 6-item Kessler Psychological Distress Scale; FDNPP: Fukushima Daiichi Nuclear Power Plant. *: significant difference using the logistic regression analysis.

## Data Availability

All data are available from the corresponding author upon reasonable request.

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
