# Peer review of "Assessment of Radiation Risk Perception and Interest in Tritiated Water among Returnees to and Evacuees from Tomioka Town within 20 km of the Fukushima Daiichi Nuclear Power Plant"

_ijerph, 2023, doi:10.3390/ijerph20032690_

Round 1

Reviewer 1 Report

Thank you for the opportunity to read and review your paper. The authors use questionnaire data to assess radiation risk perception and interest in tritiated water among people impacted by evacuation orders after the Fukushima nuclear disaster. Understanding how affected populations perceive risk is important and the comparison between evacuees’ and returnees’ perception of risk is insightful, but I have some concerns that should be addressed before publication. 

-       One of main issues lies with the fact that, as presented, the study’s contribution to the field remains limited. The paper uses other studies to situate its findings but does not provide an in-depth discussion of the strengths of the results. I suggest that the discussion and conclusions provide a more robust analysis of the findings.

-       The paper mentions “adequate” risk perceptions but does not explain that it entails in this context. I recommend providing a brief overview of risk perceptions and their importance in risk communication as well as a description of adequate risk perception.

-       The section about mental health and life satisfaction is interesting but the item measuring life satisfaction is seemingly very board. The authors could better explain how it relates to radiation anxiety and risk perception. Additionally, the inclusion of people’s interest in tritiated water is minimal and could be further developed – more analysis and discussion are derived from the age, gender, or presence of children among participants.

-        Another issue comes from the rather cumbersome writing style which, at times, makes the paper confusing – despite an overall clear argument. For instance, the interpretation of odds ratio (lines 172-181) is difficult to read. A few long sentences or phrases could be edited for clarity throughout the paper – lines 255-257, “was no significant differences” in the abstract and on line 151, repetitive statements (lines 85-88) for example, etc.

-        I encourage the authors to move the group overview included at the beginning of the “Results” section (lines 130-133) to the “Participants” section (near line 90) to give readers a better understanding of the sample and groups early on. The fact that Group 1 includes two subgroups can be confusing in the analysis – lines 135-141 for instance. Double-checking for these throughout the paper would strengthen the argument.  

Reviewer 2 Report

This paper is a highly relevant study that deserves to be published.

In the first place, it is a methodologically rigorous investigation (h in its design, in its statistical analysis and the interpretation of its results). Secondly, the relevance of their findings transcends the scope of the Fukushima Daiichi nuclear catastrophe and provides valuable and generalizable insights (beyond the singular case of the disaster and the drama and its impact on the community), regarding communication and assumption of nuclear risk. Findings that, for example, reveal the relationship between people living close to nuclear sources used to obtain information to the risks; or the relationship between risk perception and confidence in the source of information.

As a minor objection, it would perhaps have been relevant to include aspects previously outlined in the discussion in the introduction. However, the relevance of the study derived from the singularity of the case minimizes this objection.

Reviewer 3 Report

Goal of the survey is clear. The research performed is according to a fully acceptable quality level. Structure of the paper is alright.

Abstract:

Check 'differences' line 21.

Introduction:

Not much has been revealed of previous studies. Most of that is mentioned in Section 5 Discussion.

Section 2.3, lines 119-121: the sentence is not right at 'and' in the middle. Should it be two sentences? And how is group 1 defined?

Line 124: What are Groups 1 and 2?

Section 3: Table 1: Age: the second age group is that people older than 60 but younger than 70, and people of 70 and older? It is not clear. 

Lines 172-181: The text above Table 2 is not attractive to read. Please improve by introducing some order, e.g., by introducing separate paragraphs.

Section 4.1, lines 202-204: It would be interesting for readers to know the current radiation levels at Tomioka as compared to the acceptance standard level. Because if the hazard level is acceptable, then better risk communication would be justified.

Section 4.3, line 229: What 'rumors'?
